# Design and Calibration of Robot Base Force/Torque Sensors and Their Application to Non-Collocated Admittance Control for Automated Tool Changing

**DOI:** 10.3390/s21092895

**Published:** 2021-04-21

**Authors:** Hubert Gattringer, Andreas Müller, Philip Hoermandinger

**Affiliations:** Institute of Robotics, Johannes Kepler University Linz, 4040 Linz, Austria; hubert.gattringer@jku.at (H.G.); philip.hoermi@gmail.com (P.H.)

**Keywords:** force/torque sensor, sensor design, admittance control, sensor calibration

## Abstract

Robotic manipulators physically interacting with their environment must be able to measure contact forces/torques. The standard approach to this end is attaching force/torque sensors directly at the end-effector (EE). This provides accurate measurements, but at a significant cost. Indirect measurement of the EE-loads by means of torque sensors at the actuated joint of a robot is an alternative, in particular for series-elastic actuators, but requires dedicated robot designs and significantly increases costs. In this paper, two alternative sensor concept for indirect measurement of EE-loads are presented. Both sensors are located at the robot base. The first sensor design involves three load cells on which the robot is mounted. The second concept consists of a steel plate with four spokes, at which it is suspended. At each spoke, strain gauges are attached to measure the local deformation, which is related to the load at the sensor plate (resembling the main principle of a force/torque sensor). Inferring the EE-load from the so determined base wrench necessitates a dynamic model of the robot, which accounts for the static as well as dynamic loads. A prototype implementation of both concepts is reported. Special attention is given to the model-based calibration, which is crucial for these indirect measurement concepts. Experimental results are shown when the novel sensors are employed for a tool changing task, which to some extend resembles the well-known peg-in-the-hole problem.

## 1. Introduction

A novel force/torque sensor is proposed that allows measuring the ground reaction wrench at the base of a robot. The application scenario that motivated developing alternative sensor concepts and the research reported in this paper is the robotized tool change. Different tools with weight ranging from 2 to 5 kg are to be inserted into the clamping unit of a toolholder, as shown in Figure 1. A robot equipped with a dedicated gripper is used to this end. Inserting the tool in the clamping unit is the critical part of the tool change procedure. It resembles a mating process, and is in many respects similar to the classical peg-in-the-hole problem. The fundamental issue encountered are the uncertainties in the geometry and location of the toolholder, the tool itself, and the gripper, but also due to the positioning accuracy of the robot. The principle challenge therefore is that a position controlled robot is not able to execute the tool changing task as it is unable to perfectly locate the tool relative to the clamping unit, which leads to collision of robot/tool and toolholder, and consequently damages the robotic system. One way to potentially avoid such collisions is to increase the absolute positioning accuracy of the robot. This can in principle be achieved with a geometric calibration of the robot. However, the possible accuracy improvement is limited due to the inherent limitations of any robotic manipulator. The relative positioning accuracy of tool and toolholder can further be increased by means of external measurements. The EE motion and the toolholder location could be measured with laser tracking system, for instance. This is not a practically feasible solution, however. Visual servoing using a camera system represents an alternative, which is also not a practically reliable solution. The most robust approach is to control the interaction forces and to exploit them in order to navigate the relative motion. This necessitates a dynamic model of the robot as well as a geometric model of the contact situation. Moreover, a reliable force/torque measurement at the robot EE is crucial.

The de facto standard method to measure the interaction wrench at the EE is to use force/torque sensors. Application of such sensors is accompanied with several restrictions and challenges. The main issue, particularly relevant for the targeted application, is the increase of the occupied space due to the size of the force/torque sensors. The size of the sensor also leads to a change of the point were the contact force is applied at the robot. This is a problematic issue for wrist-partitioned robots, since the distal point of application increases the torque the wrist has to sustain. Beside the high cost of the force/torque sensor (which is critical when aiming at series production), another difficulty when using the sensors located at the EE is the wiring along robot links and joints.

Various alternative approaches were proposed. An alternative to measuring the interaction wrench directly at the EE, is to measure joint torques. Direct measurement of these torques demands special robot designs with direct joint torque sensors [1,2], which makes such solutions significantly more expensive than using force/torque sensors attached at the EE. In order to avoid expensive torque sensors (and dedicate robot design), methods were proposed to determine the joint torques from the motor torques, respectively motor currents [3]. This has been successfully applied to humanoid robots. Calculating the actual EE wrench requires a nonlinear model of the robot, where it is vital to identify its dynamic parameters [4]. Its applicability to industrial robots is limited, however, mainly due to the effect of joint friction. To account for friction, it was proposed to use a nonlinear model-based observer [5]. Due to dry friction, industrial robots exhibit a stick-slip phenomenon, which cannot be dealt with by solely measuring motor torques. A conceptually different method to indirectly measure the EE wrench consists in measuring the ground reaction at the base of the robot. The ground reaction wrench accounts for the EE load, the robot inertia, as well the dynamic loads to the robot motion. Thus, calculating the EE wrench from the ground reaction requires a dynamic model that captures the internal robot dynamics and its interaction with the environment, but does not need to capture the joint friction. Moreover, this model does not need to include detailed actuation or friction models. It was therefore employed as a method for compensating joint friction without internal joint torque sensors, and gave rise to a base sensor control (BSC) method [6,7,8]. The base wrench were measured with forces/torques within the base, similar to the measurement concept found in humanoid robots [9]. The main premise of this approach is a dynamic model with correct model parameters. On the other hand, these dynamic parameters can be identified by means of the base measurements [10]. In all publications, the base sensor for measuring the ground reactions is a six-axes force/torque sensor. This concept found industrial application in Fanuc’s collaborative CR robot series [11]. The base sensor measurements further enable the admittance/impedance control at the EE as reported in [12,13], which builds upon the compliance control of robot with elastic base with measurements at the EE [14,15]. This control concept is relevant for the application addressed in the following.

In this paper, two low-cost sensor concepts for base measurement are proposed and experimentally validated. The first concept uses three load cells integrated in the base mount of the robot. The second sensor concept consists of a four-spoke structure manufactured from a steel plate at which the robot is mounted, and which is equipped with eight strain gauges. A calibration method is developed for each of the two concepts. As for any base sensor concept, it is crucial that the calibration accounts for the dynamics of the robot, which in turn necessitates identification of the dynamic model parameters. In order to show their applicability, both sensor concepts are used within a robotized tool changing scenario, which relies on force measurement at the EE. The experimental results show that both sensor concepts are applicable.

## 2. Testbed for Tool Changing Application

The setup of the robot system used for the tool changing application is shown in Figure 1. Its core element was the six axes Stäubli Tx90L robot with a working range of approximately 1.2 m at a maximum payload of 10 kg. Its configuration was defined by the six joint angles $q^{T}=\left(q_{1},..,q_{6}\right)$. The novel force/torque sensors are located at the base of the robot. This setup allowed use of both of the proposed sensors. A six-axes force/torque sensor (ATI-Delta, calibration SI-660-60) and a dedicated gripper were mounted at the end-effector (EE) of the robot, located after the sixth axis. The ATI force/torque sensor was used for validation of the proposed base sensors. The tool changing scenario was tested with a specially designed tool and a stationary clamping unit that held the tool for a further work task.

The complete real-time control and sensor evaluation was carried out on a B&R Automation PC [16] connected to servo drives and input modules, running at a cycle time of 400 μs. This systems allowed for implementation of tailored control methods using Matlab/Simulink code generation, which was used to implement the model-based control. An admittance control and a position control were used as described in Section 6.

## 3. Sensor Concepts

### 3.1. Base Force/Torque Measurement with Load Cells

The robot was mounted on three load cells as shown in Figure 2, from which the wrench at the base was deduced. The static base load was determined by the 120 kg weight of the robot. The dimensioning of the load cells was based on a dynamic simulation of exemplary dynamically demanding motions using the complete nonlinear robot model. From these simulations, the peak forces at the sensors were determined, which indicated the rated load equivalent to 300 kg/sensor. Thereupon, the load cell PSD-S1 was selected with specification according to Table 1.

The cost of each load cell is 35 € so that this three-cell sensor system would allow cost-efficient base force measurement. It must be noticed, however, that the load cells are typically rather sensitive to transverse forces (orthogonal to measuring direction). For the intended application, this is particularly critical since horizontal forces will occur due to the contact. This issue is mitigated by the admittance control, which is used to limit the contact forces and thus the lateral forces.

### 3.2. Base Force/Torque Measurement with a Dedicated Sensor

A force/torque sensor was designed consisting of a four-spoke structure made from a steel plate. At each spoke a strain gauge was affixed in order to measure the local bending of the structure. The principle geometry is shown in Figure 3 and Figure 4. This concept has been used for designing force/torque sensors attached at the feet of two-legged walking machines [17]. It is a simple and cost-efficient method for measuring ground contact forces/torques. In [18], for instance, a sensor was developed using a three-spoke structure.

The specific layout must be optimized according to the particular robot. For the considered application and setup (Section 2), the thickness of the mount plate was optimized. To this end, the sheet metal with thickness of 8 mm and 10 mm were compared. For both, a finite element analysis was carried out. Three static load cases were considered (Table 2). The first load case took into account the weight of the robot and the gripper, which was estimated as 120 kg. For the FE analysis an additional safety margin of 1500 N was added to the corresponding load. The second load case, in addition to the weight force, took into account the torque generated by the robot’s own weight and the EE load. The pose that led to an extreme torque at base was attained when the arm segments were stretched out horizontally. In this pose, the EE was 1.35 m away from mount center. The maximal force at the EE of 70 N led to a torque of 94.5 Nm. In this pose, the horizontal position of the center of mass (COM) of the robot, was estimated to be 0.5 m away from the base center. The base torque due to the gravity force was almost 600 Nm. Thus, the maximal net torque at the base was estimated with 700 Nm. Figure 3 shows the corresponding displacements.

Since the tool changing application was a quasi-static process, these static load cases were sufficient. The static analysis using the FE software Solidworks investigated the von Mises stress to ensure mechanical strength of the plate, and the displacement (Figure 3) and curvature in order to select strain gauges. Both thicknesses proved to be applicable, but in order to increase the sensor sensitivity, the sheet metal with 8 mm thickness was selected. Strain gauges of type HBM 1-DY11-3/350 were chosen. The cost for 10 of these sensors was 240 €.

The base sensor structure was laser cut from an steel plate. Eight strain gauges were attached (glued) at the position that exhibited maximal strain. The latter were located at the narrowing of the plate as shown in Figure 4.

## 4. Sensor Calibration

The base sensors were used to measure the wrench at a reference point on the mounting plate. Figure 5 shows the footprint of the robot and the reference frame $\left(B\right)$ introduced at the reference point. For the considered task, the vertical force and the two horizontal components of the torque were necessary only. Denote with $h_{B}$ a wrench expressed in the base-frame $\left(B\right)$, then these three components can be extracted using a selection matrix S as
(1)$$\underset{h_{B,red}}{\underset{︸}{{\left(\begin{array}{c} f_{Bz}\;M_{Bx}\;M_{By} \end{array}\right)}^{T}}}=\underset{S}{\underset{︸}{\left[\begin{array}{cccccc} 0 & 0 & 1 & 0 & 0 & 0\\ 0 & 0 & 0 & 1 & 0 & 0\\ 0 & 0 & 0 & 0 & 1 & 0 \end{array}\right]}}\underset{h_{B}}{\underset{︸}{{\left(\begin{array}{c} f_{Bx}\;f_{By}\;f_{Bz}\;M_{Bx}\;M_{By}\;M_{Bz} \end{array}\right)}^{T}}}.$$

Now the calibration amounts to relate the output of the load cells and the strain gauges, respectively, to the reduced vector $h_{B,red}$.

### 4.1. Base Force/Torque Measurement with Load Cells

The calibration of this base sensor concept cannot be achieved by separate calibration of the individual load cells, which is due to the coupling of the three sensors and the effect of the robot. With the geometric parameters shown in Figure 5, the base wrench $h_{B,red}$ can be calculated from the forces at the three measured by the three sensors forces as
(2)$$h_{B,red}=\left(\begin{array}{c} f_{Bz}\\ M_{Bx}\\ M_{By} \end{array}\right)=\underset{\Theta_{1}}{\underset{︸}{\left[\begin{array}{ccc} -1 & -1\; & -1\\ 0 & -\frac{b_{B}}{2} & \frac{b_{B}}{2}\\ l_{B,1} & -l_{B,2} & -l_{B,2} \end{array}\right]}}\left(\begin{array}{c} f_{w1}\\ f_{w2}\\ f_{w3} \end{array}\right).$$

Each load cell is part of a measuring bridge with output voltage $u_{wi},i=1,2,3$. An affine linear relation of these voltages and the forces at the load cells is assumed as follows
(3)$$f_{wi}=k_{wi}\;u_{wi}+d_{wi},\;\;\;\;i=1,2,3$$
where $d_{wi}$ is a force offset. This can be written in the form
(4)$$\left(\begin{array}{c} f_{w1}\\ f_{w2}\\ f_{w3} \end{array}\right)=\underset{\Theta_{2}}{\underset{︸}{\left[\begin{array}{cccccc} u_{w1} & 1 & 0 & 0\; & 0 & 0\\ 0 & 0 & u_{w2} & 1 & 0 & 0\\ 0 & 0 & 0 & 0 & u_{w3} & 1 \end{array}\right]}}\underset{p_{w}}{\underset{︸}{\left(\begin{array}{c} k_{w1}\\ d_{w1}\\ k_{w2}\\ d_{w2}\\ k_{w3}\\ d_{w3} \end{array}\right)}}.$$

The vector $p_{w}$ comprises the parameters in the linear model to be determined. Combining (Equation 2) and (Equation 4) yields the overall relation in identification form
(5)$$h_{B,red}=\Theta_{1}\Theta_{2}p_{w}.$$

For given voltages, (Equation 5) allows to relate parameters of the sensor model to the base wrench. In order to determine the parameters for given base wrench, a series of *N* measurements is carried out. Denote with $\Theta_{w,j}:=\Theta_{1}\Theta_{2,j}$ the coefficient matrix in (Equation 5) evaluated with the voltages of measurement j=1,…,N, and the corresponding base wrenches with $h_{B,red,j}$. Then the final identification form is
(6)$$Q_{w}=\Theta_{w}p_{w}$$
with
(7)$$\Theta_{w}=\left(\begin{array}{c} \Theta_{w,1}\\ \vdots \\ \Theta_{w,N} \end{array}\right),\;\;Q_{w}=\left(\begin{array}{c} h_{B,red,1}\\ \vdots \\ h_{B,red,N} \end{array}\right).$$

The least squares solution minimizing $∥Q_{w}-\Theta_{w}p_{w}∥$ is
(8)$$p_{w}={\left(\Theta_{w}^{T}\Theta_{w}\right)}^{-1}\Theta_{w}^{T}Q_{w}.$$

### 4.2. Base Force/Torque Measurement with a Dedicated Sensor

Each pair of strain gauges is included in a measuring bridge, which outputs a voltage denoted $u_{mi},i=1,2,3,4$. The voltage measurements are summarized in the vector
(9)$$u_{m}=\left(\begin{array}{c} u_{m1}\\ u_{m2}\\ u_{m3}\\ u_{m4} \end{array}\right).$$

An affine linear sensor model is introduced as
(10)$$h_{B,red}=S_{m}\left(u_{m}-u_{m,off}\right)=S_{m}u_{m}-h_{B,red,off}=\left(\begin{array}{c} S_{m,1}u_{m}\\ S_{m,2}u_{m}\\ S_{m,3}u_{m} \end{array}\right)-\left(\begin{array}{c} f_{Bz,off}\\ M_{Bx,off}\\ M_{By,off} \end{array}\right)$$
with sensor matrix
(11)$$S_{m}=\left[\begin{array}{cccc} S_{m,11} & S_{m,12} & S_{m,13} & S_{m,14}\\ S_{m,21} & S_{m,22} & S_{m,23} & S_{m,24}\\ S_{m,31} & S_{m,32} & S_{m,33} & S_{m,34} \end{array}\right]=\left(\begin{array}{c} S_{m,1}\\ S_{m,2}\\ S_{m,3} \end{array}\right).$$

The relation (Equation 10) is linear in the model parameters $S_{m}$ and $h_{B,red,off}$, and for each measurement j=1,…,N, it can be written as
(12)$$h_{B,red,j}=\underset{\Theta_{m,j}}{\underset{︸}{\left[\begin{array}{cccccc} u_{m}^{T} & 1 & 0 & 0 & 0 & 0\\ 0 & 0 & u_{m}^{T} & 1 & 0 & 0\\ 0 & 0 & 0 & 0 & u_{m}^{T} & 1 \end{array}\right]}}\underset{p_{m}}{\underset{︸}{\left(\begin{array}{c} S_{m,1}^{T}\\ f_{Bz,off}\\ S_{m,2}^{T}\\ M_{Bx,off}\\ S_{m,3}^{T}\\ M_{By,off} \end{array}\right)}}.$$

In analogy to (Equation 6), these relation are summarized as
(13)$$Q_{m}=\Theta_{m}p_{m}.$$

A solution of this linear regression problem is again obtained by the least squares solution
(14)$$p_{m}={\left(\Theta_{m}^{T}\Theta_{m}\right)}^{-1}\Theta_{m}^{T}Q_{m}.$$

### 4.3. Calculation of Base Wrench for Calibration

The calibration Equations (Equation 8) and (Equation 14) contain the respective voltages but also the base wrench. The latter is not known explicitly, and must be computed from the motor torques and the robot inertia. Denote with $M_{mot,2}$ the drive torques of the second motor (see Figure 6). Then the effective torque acting on the link is $i_{G,2}M_{mot,2}$, where $i_{G,2}$ is the gear ration of the motor-gear unit. Because of the asymmetric layout of the robot, there is a torque $M_{x,off}$ as shown in Figure 6. This cannot be measured, and is regarded as an offset torque. Transforming the overall torque to the base frame yields the wrench
(15)$$h_{B,red}^{*}=\left(\begin{array}{c} f_{Bz}\\ M_{Bx}\\ M_{By} \end{array}\right)=\left(\begin{array}{c} -m_{rob}\;g\\ -\sin\left(q_{1}\right)i_{G,2}M_{mot,2}+\cos\left(q_{1}\right)M_{x,off}\\ \cos\left(q_{1}\right)i_{G,2}M_{mot,2}+\sin\left(q_{1}\right)M_{x,off} \end{array}\right)$$
where $q_{1}$ denotes the joint angle of the first robot axis, $m_{rob}$ is the total mass of the robot, and *g* is the gravitational constant.

In a given pose j=1,…,N, according to $q_{1,j}$ and with motor torque $M_{mot,2,j}$, relation (Equation 15) can be incorporated into the identification form (Equation 5) or (Equation 13), which leads to
(16)$$\underset{{\bar{Q}}_{j}}{\underset{︸}{\left(\begin{array}{c} -m_{rob}\;g\\ -\sin\left(q_{1,j}\right)i_{G,2}M_{mot,2,j}\\ \cos\left(q_{1,j}\right)i_{G,2}M_{mot,2,k} \end{array}\right)}}=\underset{{\bar{\Theta }}_{j}}{\underset{︸}{\left[\begin{array}{cc} \Theta_{j} & |-\left(\begin{array}{c} 0\\ \cos\left(q_{1,j}\right)\\ \sin\left(q_{1,j}\right) \end{array}\right) \end{array}\right]}}\underset{\bar{p}}{\underset{︸}{\left(\begin{array}{c} p\\ M_{x,off} \end{array}\right)}}.$$

The final set of calibration equations is thus obtained as
(17)$$\bar{Q}=\bar{\Theta }\bar{p}$$
with $\bar{Q}$ and $\bar{\Theta }$ comprising ${\bar{Q}}_{j}$ and ${\bar{\Theta }}_{j}$, respectively, as in (Equation 7).

### 4.4. Calibration Procedure and Results

It was important to select calibration poses to cover the relevant workspace. Therefore, the *N* measurements were carried out for robot configurations obtained by varying the first two joints with joint coordinates $q_{1}$ and $q_{2}$, respectively. All other joint coordinates were set to zero, which corresponds to reference configuration in Figure 6 (right). Measurements were obtained for the following values of the first joint coordinate: $q_{1}=-135{}^{\circ },-90{}^{\circ },-45{}^{\circ },0{}^{\circ },45{}^{\circ },90{}^{\circ },135{}^{\circ }$ (see Figure 7). For each value of $q_{1}$, the second coordinate $q_{2}$ is varied starting from $90{}^{\circ }$ to $-5{}^{\circ }$, in steps of $5{}^{\circ }$. As representative example, Figure 8 shows the measured voltages and the motor torques when $q_{1}=-45^{\circ }$ and $q_{2}$ was set according the above values. With the so obtained calibration points (indicated by circles in Figure 8), the calibration was pursued with solution (Equation 8) in order to identify the parameters of the three load cell sensor system, and the solution (Equation 14) for parameter identification of the four-spoke sensor.

Using the identified sensor parameters, the accuracy of the estimation from either sensor concept could be compared from base wrenches when calculated with (Equation 15). Figure 9 shows the difference $e_{f}:=f_{Bz}-f_{Bz}^{*}$ of the estimated force and the differences $e_{M,x}:=M_{Bx}-M_{Bx}^{*},e_{M,y}:=M_{By}-M_{By}^{*}$ of the estimated torque for the four-spoke sensor. The load cell sensor was of similar quality.

### 4.5. Compensation of Robot Dynamics

The base measurement captured the static load according to the current pose of the robot, the EE loads, but also the dynamic loads due to the robot motion. For the tool change application, the aim was to measure the applied EE load only, which was used with the admittance control. Consequently, the static and dynamic load due to robot must be eliminated from the base measurement.

The wrench due to the robot was determined from the dynamic robot model as
(18)$$h_{B,rob}=f\left(q,\dot{q},\ddot{q},p_{rob}\right)$$
where $p_{rob}$ is the vector with dynamic model parameters. A systematic way to evaluate this model can be found in [19]. The net base wrench attributed to the applied external EE load was then estimated as
(19)$$h_{B,ext,red}=h_{B,red}-h_{B,rob,red}.$$

When evaluating (Equation 19), it is crucial that the dynamic parameters in $p_{rob}$ are known exactly. Identification methods were developed to this end [4]. This was performed for the Stäubli Tx90L robot used within the testbed. The problematic point, however, is that not all dynamic parameters are independent and thus only the so-called base parameters can be identified. This is not a problem when solving the inverse dynamics, i.e., computing the joint torques for given motion, but it is critical when evaluating (Equation 18), since the ground reaction wrenches depend on the complete set of dynamic robot parameters if the robot is moving. On the other hand, in the tool changing application, the robot performs quasi-static motions only. Moreover the orientation of the robot links remains almost constant. This suggests introducing a static model that linearly relates the EE-position and the base wrench
(20)$$\underset{Q_{S,j}}{\underset{︸}{h_{B,red}}}=\underset{\Theta_{S,j}}{\underset{︸}{\left(\begin{array}{ccc} {}_{I}r_{BE}^{T} & 0 & 0\\ 0 & {}_{I}r_{BE}^{T} & 0\\ 0 & 0 & {}_{I}r_{BE}^{T} \end{array}\right)}}\underset{p_{S}}{\underset{︸}{\left(\begin{array}{c} S_{1}^{T}\\ S_{2}^{T}\\ S_{3}^{T} \end{array}\right)}}$$
where $S_{i}$ represents a stiffness matrix 6×3 collocated at the EE, and associated to sensor *i*, such that $h_{B,red}=S_{i}{}_{I}r_{BE}$.

A series of j=1,…,m measurements is performed for the motion along the EE-path which is followed during the tool changing task. The measured base forces/torque are shown in Figure 10 left. Each measurement of $h_{B,red}$ and ${}_{I}r_{BE}$ gives rise to $\Theta_{S,j}$ and $Q_{S,j}$, as indicated in (Equation 20). Collecting these in $\Theta_{S}$ and $Q_{S}$ yields the linear regression problem $Q_{S}=\Theta_{S}p_{S}$. Its least squares solution is
(21)$$p_{S}={\left(\Theta_{S}^{T}\Theta_{S}\right)}^{-1}\Theta_{S}^{T}Q_{S}.$$

Figure 10 shows the base forces/torque computed with the linear model (Equation 20) for a quasi-static motion along the tool changing path, where the tool/EE does not contact the toolholder. For this application, the model yields sufficiently accurate estimate of the base forces/torque.

## 5. Computation of End-Effector Wrenches

The overall goal was to deduce the applied wrench at the EE. The wrench applied at the EE and the base wrench were related by the transformation
(22)$$\underset{{}_{I}h_{B}}{\underset{︸}{\left(\begin{array}{c} {}_{I}f_{B}\\ {}_{I}M_{B} \end{array}\right)}}=\underset{T_{BE}}{\underset{︸}{\left[\begin{array}{cc} I\; & 0\\ {}_{I}{\tilde{r}}_{BE} & I \end{array}\right]\left[\begin{array}{cc} R_{I6} & 0\\ 0 & R_{I6} \end{array}\right]}}\underset{{}_{6}h_{E}}{\underset{︸}{\left(\begin{array}{c} {}_{6}f_{E}\\ {}_{6}M_{E} \end{array}\right)}}$$
where ${}_{6}h_{E}$ is the EE wrench represented in the body-fixed frame at the last (sixth) link, ${}_{I}h_{B}$ is the wrench at the ground represented in the global inertial frame $\left(I\right)$, and ${}_{I}r_{BE}$ is the position vector from the base frame to the EE frame represented in the inertial frame (see Figure 11). The reduced base wrench comprising the measurements ${}_{I}f_{Bz}$, ${}_{I}M_{Bx}$, ${}_{I}M_{By}$ was thus determined by
(23)$${}_{I}h_{B,red}=\left(\begin{array}{c} {}_{I}f_{Bz}\\ {}_{I}M_{Bx}\\ {}_{I}M_{By} \end{array}\right)=\underset{T_{BE,red}}{\underset{︸}{ST_{BE}}}{{}_{I}h}_{E}.$$

This is an overdetermined linear system in the unknown EE wrench. The latter can be determined by solving the minimization problem
(24)$$\begin{array}{ccc} & & \min_{{}_{I}h_{E}}\frac{1}{2}{{}_{I}h}_{E}^{T}W\;{{}_{I}h}_{E}\\ & & s.t.\;{\;}_{I}h_{B,red}=T_{BE,red}\;{{}_{I}h}_{E} \end{array}$$
where W is a weighting matrix. It has the unique solution
(25)$$\begin{array}{ccc} {{}_{I}h}_{E} & = & W^{-1}T_{BE,red}^{T}{\left(T_{BE,red}W^{-1}T_{BE,red}^{T}\right)}^{-1}{{}_{I}h}_{B,red} \end{array}$$
(26)$$\begin{array}{ccc} & = & T_{BE,red}^{\#}{{}_{I}h}_{B,red}. \end{array}$$

For the tool changing task, the contact forces at the EE were most relevant, the weighting matrix could be chosen as
(27)W=diag[1,1,1,50,50,50].

This forces could be directly used for admittance control.

## 6. Admittance Control

The tool changing task was executed by controlling the contact force between the tool and the toolholder. An admittance control scheme was used to this end, which indeed relied on a sufficiently accurate measurements of the contact force. The EE forces estimated with the two sensor concepts provided such a reliable measurement. The dynamic equations of motion of the robot formed the basis for the admittance controller. They could be expressed in joint coordinates as
(28)$$M\ddot{q}+g=Q_{M}-J^{T}h_{E,ext}$$
where M is the generalized mass matrix, g represents all generalized forces not depending on the acceleration, and and $Q_{M}$ is the vector of motor torques [20,21]. Further J, is the manipulator Jacobian, which determines the EE twist $V_{E}$ in terms of the joint velocities as
(29)$$V_{E}=J\dot{q},$$
and $h_{E,ext}$ is the wrench at the EE due to external loads, e.g., contact forces. For the non-redundant 6-DOF manipulator, the Jacobian is an invertible square matrix, so that the motion equations can be transformed into task space. The task space formulation of the equations of motion is
(30)$$\Lambda_{E}{\dot{V}}_{E}+J^{-T}g-\Lambda_{E}\dot{J}\dot{q}=J^{-T}Q_{M}-h_{E,ext}$$
where the mass matrix in operational space is defined as
(31)$$\Lambda_{E}:={\left(JM^{-1}J^{T}\right)}^{-1}.$$

An admittance control scheme could then be introduced as [20]
(32)$$\Lambda_{d}\Delta {\dot{V}}_{E}+D_{d}\Delta V_{E}+K_{d}\Delta z_{E}=h_{E,ext}.$$
Here $\Delta z_{E}$ is the difference of the current and the desired EE coordinates, and $\Delta V_{E}:=V_{E,d}-V_{E}$ is the different of EE twist. The system (Equation 32) describes the desired dynamics, i.e., the compliant behavior of the robot. According to the EE load $h_{E,ext}$, the EE moves similar to an elastic system with dynamics governed by (Equation 32). The schematic setup of the admittance control scheme is shown in Figure 12. Further details on the control concept can be found in [20,21].

## 7. Application to Automatic Tool Changing

The two novel sensor concept were applied to the task of changing a tool mounted in a toolholder, and the performance of the calibrated sensors was investigated in detail. From the sensor perspective, this task could be approached from two different directions. One was relying on the available intrinsic sensor information, which usually meant encoder data to infer the joint rotations and thus the robot pose. Only dedicated robots were equipped with integrated joint torque sensors (e.g., robots intended for human robot interaction). Consequently, an open-loop control required highly precise absolute positioning in order to insert the tool into the holder. This, however, was not possible with standard industrial robots, even if geometrically calibrated. The second approach was to use extrinsic sensors, which provided visual information about the scene or determine the interaction force/torque when the robot was in contact with its environment. For applications resembling the ‘peg-in-the-hole’ task, the accuracy of visual data (e.g., 3D point clouds, 2D image fusion) is usually not sufficient to accurately guide the EE. Force measurements, on the other hand, do provide the tactile feedback to robustly navigate the EE during the tool changing operation. The global positioning toward the toolholder is hence safely achieved using the robot position control.

The navigation during the tool changing process therefore relied on a force control at the robot EE by means of an admittance controller. The two sensor concepts were employed. The obtained results are described in the following.

### 7.1. Tool Changing Procedure

The tool changing task was to grasp and remove the tool currently fixed at toolholder, and to insert and release another tool into the holder. In order to test the procedure the following sequence of subtasks was performed:Approach tool and get gripper into contact with toolEngage tool by gripperUnlock tool from toolholderLift gripper to remove tool from toolholderDisplace the tool (simulating an unknown deviation)Insert tool into toolholderLock tool in toolholderRelease tool from gripperReturn gripper to reference location

Instead of replacing the tool, task 5 served to simulate a displacement of the (second) tool that may have occurred when it was grasped by the gripper. The force control was vital for tasks 1, 4, and 6. The insertion was achieved by the geometric guidance tool relative to the hold in combination with an admittance control. The vertical position of the tool controlled by the robot. When moving downward the tool was allowed to slide down along the clamp unit (with radius chamfer) so to slip into the toolholder, where the horizontal position was dictated by the holder. The compliant horizontal motion was achieved with the admittance control, which was used to regulate the horizontal contact force. This combination of vertical position control and horizontal force control were the constitutive elements of the proposed approach while the particular contact geometry decided about the allowed horizontal deviations, respectively the misalignment of the tool when grasped by gripper. Figure 13 shows the geometry of tool and toolholder.

### 7.2. Experimental Results

Experiments were conducted as a proof of concept of the two sensor concepts and their applicability to the tool changing task. As shown in Figure 1, the robot was additionally equipped with a (ATI-Delta, calibration SI-660-60), which was mounted before the gripper. This allowed reference measurements of the contact forces/torques for comparison with the forces/torques deduced from the two proposed measurement concepts.

The above nine tasks were carried out repeatedly for 287 cycles. In subtask 5, a random displacement from the nominal location of the tool was added. It turned out that, without the admittance controller (standard position control of the robot), the maximal deviation in position and alignment for which the tool could be successfully inserted into the toolholder was ±0.25 mm and ±0.2${}^{\circ }$. With the admittance controller the maximal deviation was increased to ±1 mm, ±1${}^{\circ }$. Moreover, without the admittance control there were large lateral forces that led to undesirable load. An experiment was considered successful if the EE torque and force was below 20 Nm and 100 N, respectively. Figure 14 shows the forces/torques for three exemplary cycles when admittance control was carried out with base measurement using the load cells sensor concept (Section 3.2, Figure 4). The forces/torques were those obtained with the base load cell sensor (shown are the raw measurement data). The results obtained with the four-spoke sensor at the base were comparable. A video of the robotic tool changing experiment can be found at https://youtu.be/AVTvQgC4feQ (last accessed on 20 April 2021).

Clearly, the base force measurement provided valuable information about the ground reaction. The identified dynamic model of the robot was indeed indispensable for perusing this measurements.

## 8. Discussion and Outlook

The experimental results indicated that both sensor concepts for indirect measurements were functional and that they could be used for admittance control during quasi-static motions. The admittance control was successfully employed to limit the contact forces (in contact direction) that would otherwise make the tool changing (inserting of the tool) impossible, and would lead to unacceptable and harmful load cases of the robot. The sensor concept using the three load cells (Section 3.1) yielded the same (position and admittance) controller performance as the control with force/torque sensor at the EE. With the admittance control, the admissible deviations of the tool location and alignment was increased to ±1 mm and ±1${}^{\circ }$, respectively (compared to ±0.25 mm, ±0.2${}^{\circ }$ when only using position control of the robot). The load cell base sensor gave slightly more accurate results than the four-spoke strain gauge sensor (Section 3.2). Moreover, for more dynamic applications the reported design of the strain gauge sensor was too compliant, which would necessitate the use of a notch filter in order avoid excitation of the low eigenfrequencies. The 8 mm steel plate was chosen to provide mechanical stresses sufficient for the used strain gauges. Future work will address the optimization of the mechanical layout of this sensor and the selection of more sensitive (more expensive) strain gauges. For the robotized tool-changing application the geometry of the toolholder and the tool can be tailored by simply introducing conical shape at the opening of the toolholder.

A crucial aspect of any base sensor concept for indirect measurement of EE loads is the calibration. The latter relies on the dynamic model of the robot. As described in Section 4.5, the bottleneck is that the established dynamic identification methods [4] can only identify the so-called base parameters. Yet current research [22] addresses identification of the complete set of dynamic parameters. With these methods, the full set of parameters can be identified providing the basis for dynamic compensation using (Equation 18), and eventually making the base sensor concepts applicable to general dynamically demanding situations. Such a dynamic calibration of base sensors will be a focus of future work. Exploiting these concepts (that intrinsically introduce compliance at the base) will require advanced control and vibration damping concepts. This will be another aspect of future research, which will build upon existing approaches such as [14,15].

## 9. Conclusions

In this paper two novel model-based concepts for indirect measurement of the end-effector load of a robot are presented. The first concept employs three load cells to measure the ground reaction wrench at the base of the robot, and the second uses a tailored sensor consisting of a four-spoke plate equipped with four strain gauges resembling a classical force/torque sensor. The indirect measurement approach relies on an accurate robot model. For both concepts the corresponding calibration method is developed in the paper. A crucial aspect is the identification of the dynamic robot parameters. This is a topic of current research. This problem can be overcome for quasistatic motion scenarios, as it is shown in this paper. Experimental results for a tool changing task show that the method is applicable to non-collocated admittance control.

## Figures and Tables

**Figure 1 sensors-21-02895-f001:**
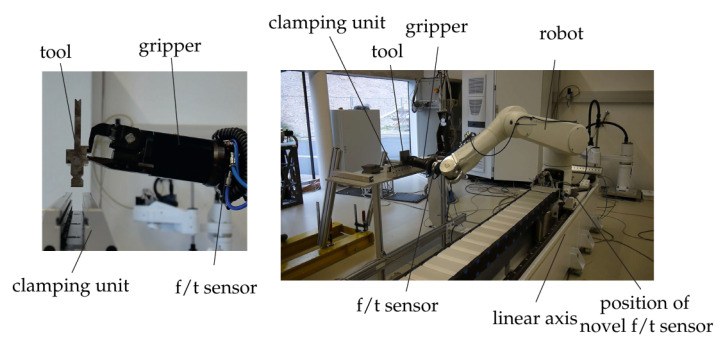
Test setup for robotic tool changing.

**Figure 2 sensors-21-02895-f002:**
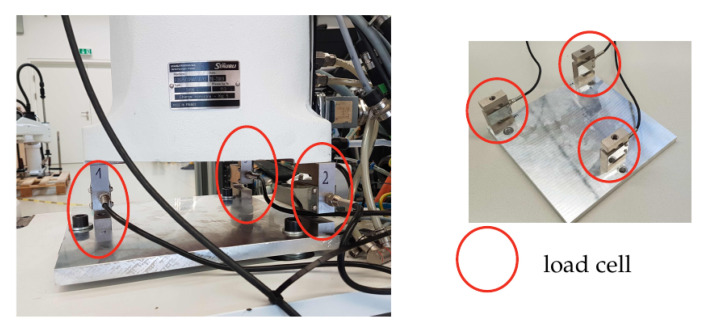
Sensor concept for measuring the base wrench using three load cells.

**Figure 3 sensors-21-02895-f003:**
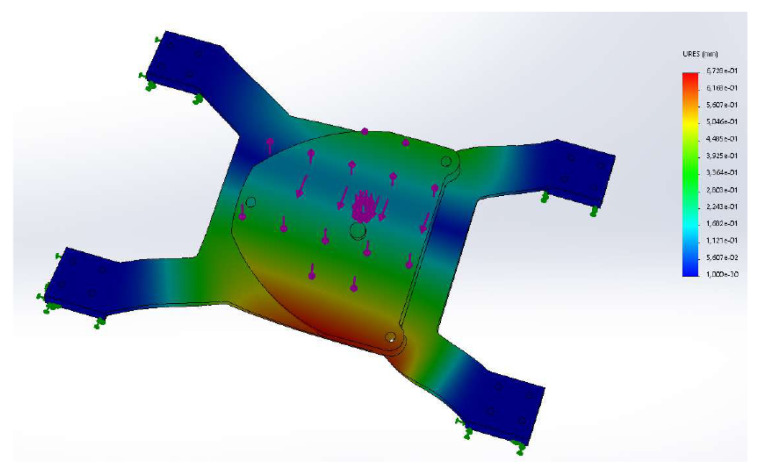
Displacement of the 8 mm steel plate obtained with a FE calculation for load case 2.

**Figure 4 sensors-21-02895-f004:**
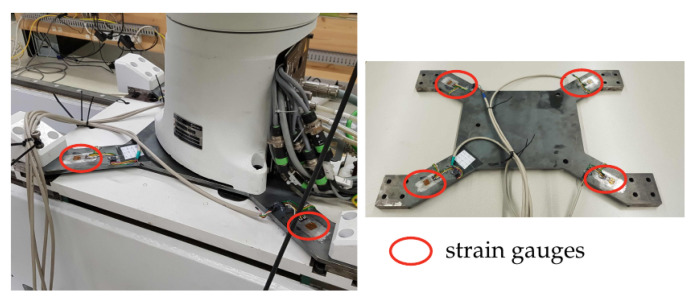
Four-spoke steel plate with strain gauges attached.

**Figure 5 sensors-21-02895-f005:**
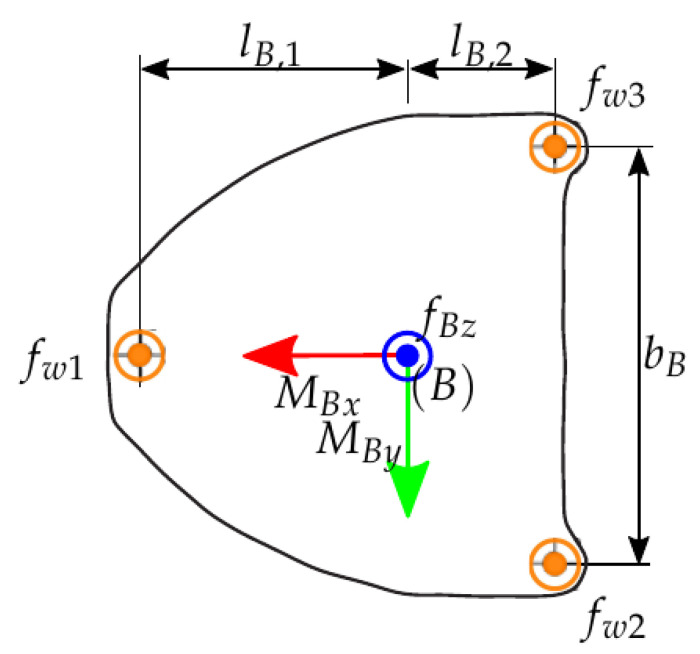
Footprint of Stäubli Tx90L robot with forces.

**Figure 6 sensors-21-02895-f006:**
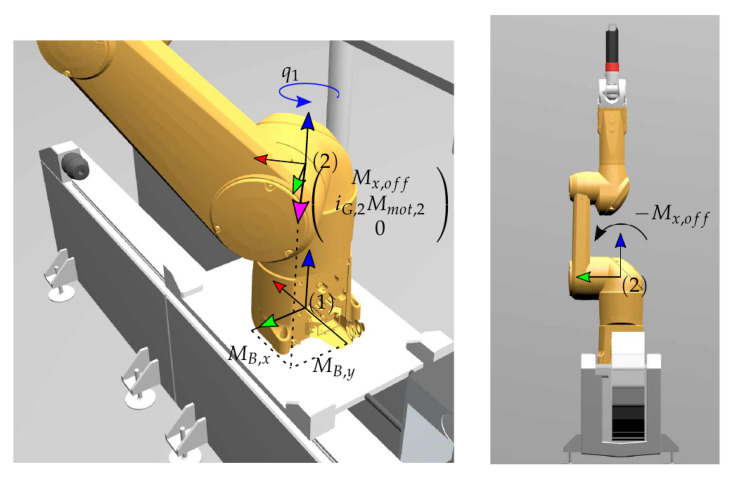
Indirect measurement of reduced base wrench and offset torque.

**Figure 7 sensors-21-02895-f007:**
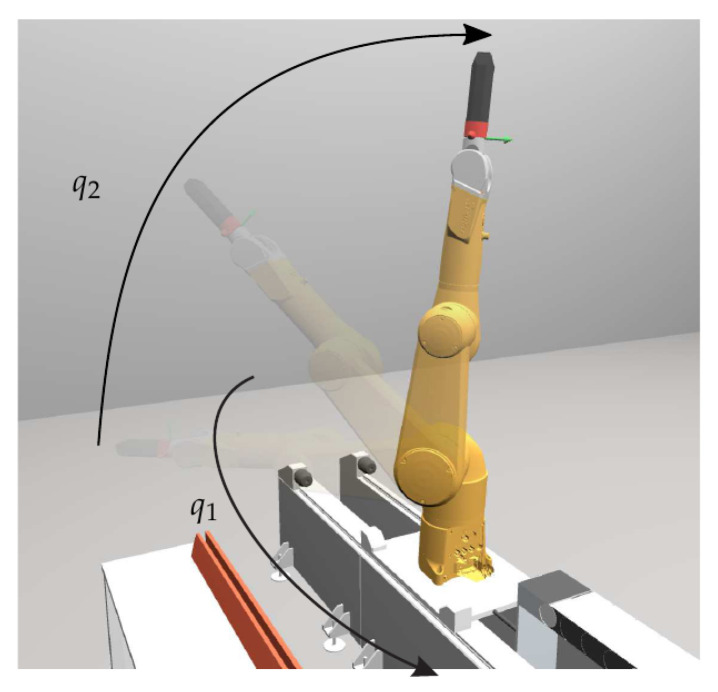
Visualization of the robot poses used for calibration.

**Figure 8 sensors-21-02895-f008:**
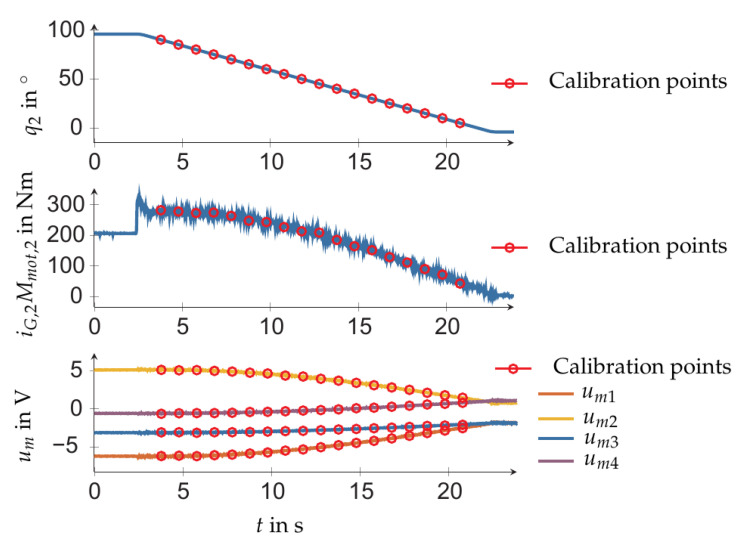
Exemplarily measurement of a calibration motion for four-spoke sensor.

**Figure 9 sensors-21-02895-f009:**
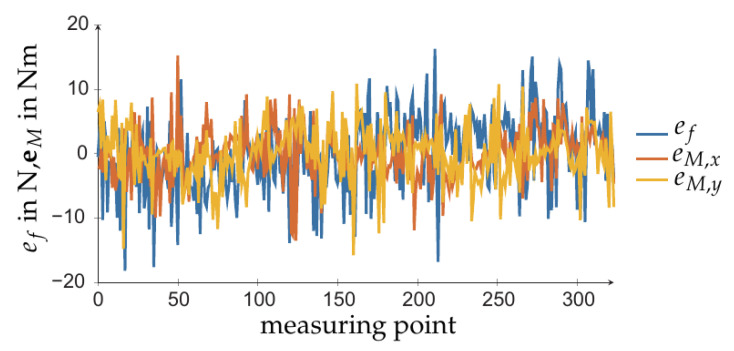
Calibration result for four-spoke sensor.

**Figure 10 sensors-21-02895-f010:**
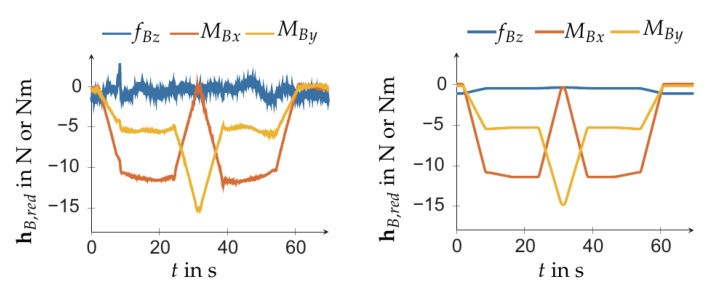
Base forces and torque during tool changing motion when the tool does not get into contact with the toolholder. Left: measured base forces/torque. Right: forces/torque computed with the linear model (Equation 20).

**Figure 11 sensors-21-02895-f011:**
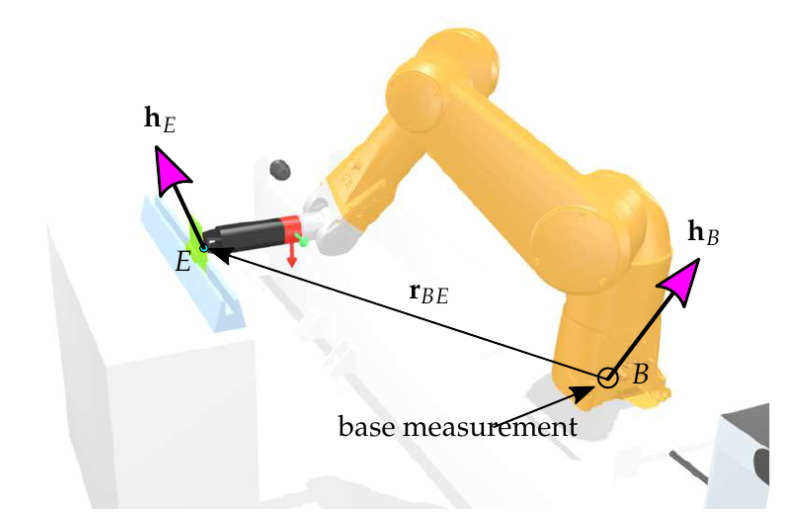
Contact scenario during tool change.

**Figure 12 sensors-21-02895-f012:**
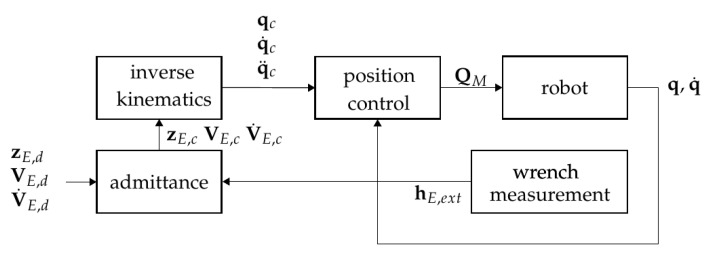
Admittance control scheme.

**Figure 13 sensors-21-02895-f013:**
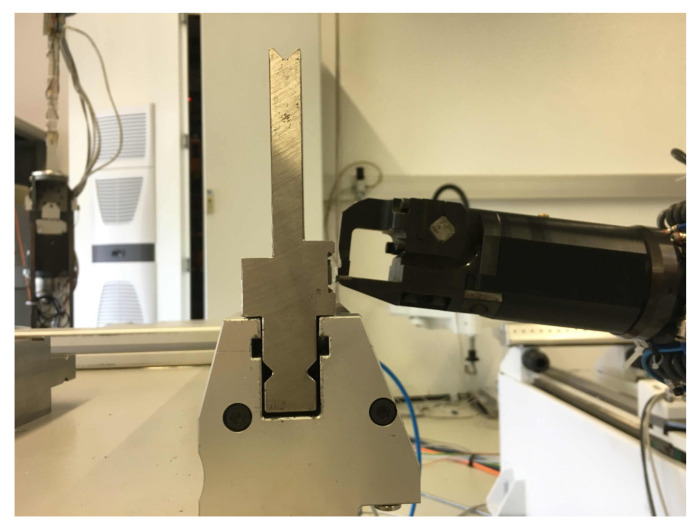
Geometry of tool and toolholder.

**Figure 14 sensors-21-02895-f014:**
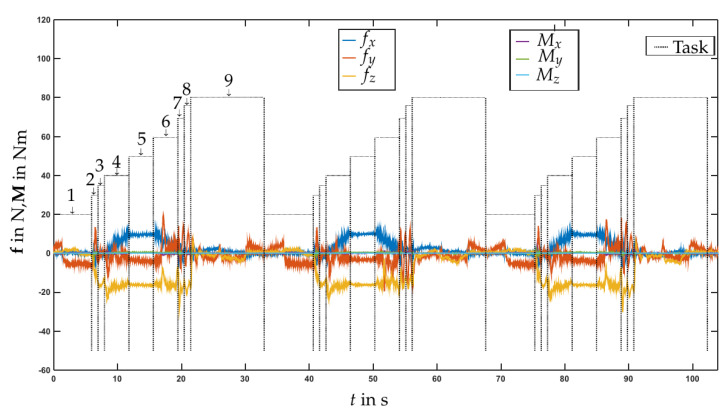
Test case tool changing.

**Table 1 sensors-21-02895-t001:** Specification of load cell *PSD-S1* according to manufacturer datasheet.

Maximum load	300 kg
Supply Voltage	10 V
Resistance	350 Ω
Sensitivity	2 millivolt per volt
Uncertainty	±0.004 millivolt per volt

**Table 2 sensors-21-02895-t002:** Load cases for FE Calculations.

Load Case	$f_{z}$ [N]	$M_{x}$ [Nm]	$M_{y}$ [Nm]
1	1500	0	0
2	1500	700	0
3	1500	0	700

## Data Availability

Not applicable.

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
