# Peer review of "Design and Calibration of Robot Base Force/Torque Sensors and Their Application to Non-Collocated Admittance Control for Automated Tool Changing"

_sensors, 2021, doi:10.3390/s21092895_

Round 1

Reviewer 1 Report

Please find the reviewer's comment as an attachment.

Author Response

We thank the reviewer for the constructive comments and remarks. The paper has been revised following the reviewer's recommendation. Please find our rebuttal letter as pdf file.

Reviewer 2 Report

The proposed paper describes two solutions for the estimation of the EE forces by using force sensors under the robot base. The proposed equipment is interesting due to its simplicity and cheapness.

However, authors should clarify some aspects prior to publication:

  • There are several typos and English errors: please double-check the entire manuscript. E.g. (not exhaustive):
    • Row 32: "increase to accuracy" > "increase the accuracy"
    • Row 57: "however, which" makes no sense
    • Row 58: "expansive" > "expensive"
    • Row 85: [14],[15] > [14,15]
  • I have some concerns regarding the solution with the three load cells: isn't the system susceptible to lateral forces? Could the robot fall due to these forces?
  • In the model dynamic forces are included, but Authors state that "the tool changing application is a quasi-static process", so "static forces are sufficient". The final impedance model used in the experiments do include the dynamic model?
  • I have some concerns regarding the strain gauge solution: in Figure 3 are shown plate displacements that are well above robot repeatability (0.6 mm vs 0.035 mm). Such displacement can greatly influence the robot performance: has it been taken into account or is it "hidden" via the calibration?
  • Between rows 196 and 197: in Equation 22 IrBE(tilde) is shown, whilst in the text rBE is cited. Please use the same notation (if they refer to the same value)
  • Has joint stiffness been considered in the model? They are not negligible as shown in [1-2]

[1] Bottin, M.; Cocuzza, S.; Comand, N.; Doria, A. Modeling and Identification of an Industrial Robot with a Selective Modal Approach. Appl. Sci. 2020, 10, 4619. https://doi.org/10.3390/app10134619

[2] Doria, A.; Cocuzza, S.; Comand, N.; Bottin, M.; Rossi, A. Analysis of the Compliance Properties of an Industrial Robot with the Mozzi Axis Approach. Robotics 2019, 8, 80. https://doi.org/10.3390/robotics8030080

Author Response

(The authors gave the same response as above.)

Round 2

Reviewer 1 Report

All comments of the reviewer are included in the revised version of the paper with the exception of the comment number 3 referring the uncertainty. In my opinion, this is an important point.

Please discuss the procedure for uncertainty determination , the value of which is included in Table 1. What is the type of uncertainty?

Author Response

Dear Reviewer,

Excuse us for not realizing what you meant in your review. 

The uncertainty mentioned in table 1 is the one listed in the data sheet of the load cell PSD-S1. To make this clear the table caption has been changed to "Specification of load cell PSD-S1 according to manufacturer datasheet".

Thank you for pointing this out.

Round 3

Reviewer 1 Report

All comments of the reviewer have been included in the revised version of the paper. I recommend publication this paper in the present form.